# Breast Glandular and Ductal Volume Changes during the Menstrual Cycle: A Study in 48 Breasts Using Ultralow-Frequency Transmitted Ultrasound Tomography/Volography

**James Wiskin [1,*] , John Klock [1] and Susan Love [2,†]**

1   QT Imaging Holdings, 3 Hamilton Landing, Ste 160, Novato, CA 94949, USA; john.klock@qtimaging.com
2   Dr. Susan Love (Deceased) Fund for Breast Cancer Research, Tower Foundation, c/o C. C. Conway, 8767 Wilshire Boulevard, Suite 401, Beverly Hills, CA 90211, USA
*   Correspondence: james.wiskin@qtimaging.com; Tel.: +1-415-842-7241
†   Deceased author.

**Abstract:** The aim of this study was to show for the first time that low-frequency 3D-transmitted ultrasound tomography (3D UT, volography) can differentiate breast tissue types using tissue properties, accurately measure glandular and ductal volumes in vivo, and measure variation over time. Data were collected for 400 QT breast scans on 24 women (ages 18–71), including four (4) postmenopausal subjects, 6–10 times over 2+ months of observation. The date of onset of menopause was noted, and the cases were further subdivided into three (3) classes: pre-, post-, and peri-menopausal. The ducts and glands were segmented using breast speed of sound, attenuation, and reflectivity images and followed over several menstrual cycles. The coefficient of variation (CoV) for *glandular tissue* in premenopausal women was significantly larger than for postmenopausal women, whereas this is not true for the *ductal* CoV. The glandular standard deviation (SD) is significantly larger in premenopausal women vs. postmenopausal women, whereas this is not true for ductal tissue. We conclude that ducts do not appreciably change over the menstrual cycle in either pre- or post-menopausal subjects, whereas glands change significantly over the cycle in pre-menopausal women, and 3D UT can differentiate ducts from glands in vivo.

**Keywords:** 3D ultrasound tomography; breast cancer; fibroglandular breast density; fibroglandular volume; ductal mapping; historically underserved populations; gland volume; menstrual cycle; glandular change; ductal change

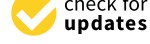



## 1. Introduction

Studies of breast anatomy are important as they enable the delineation of the normal trajectory of changes as a woman passes through puberty, menstruation, pregnancy, post-partum nursing, and menopause. In addition, for women who develop breast diseases, it is helpful to know how, where, and why breast cancer develops. Finally, breast tissue glandular and ductal volumes can be important to measure in women on various hormone treatments for benign and malignant conditions, such as benign breast and gynecological conditions, cancer prevention treatments, and treatments of hormone-receptor positive cancers with hormones or SERMs (selective estrogen receptor modulators).

There have been many treatises on breast anatomy, such as the classical book by Cooper [1] and the recent excellent review by Johnson [2]. With the advent of newer imaging technologies, there have been newer 3D volumetric studies in normal and lactating women [3–5]. These studies have enabled a more detailed view of the structure and function of the human breast.

There are also newer approaches to breast cancer management that may involve ductal approaches for diagnosis and treatment, such as ductal fluid aspiration for diagnosis and the treatment of ductal carcinomas via ductal cannulation.

The key anatomical structures in the breast include skin, fat, fascial layers, Cooper ligaments, fibroglandular tissue, lymphatics, and neurovascular structures, all positioned over the chest wall. In women, fibroglandular tissue volumes vary with age, with many women having a predominance of fat within the breasts after menopause. Normal breast anatomy can be seen using a variety of advanced imaging modalities. Tools that assess breast vascular structures (e.g., contrast-enhanced breast magnetic resonance imaging) and lymphatic structures (nuclear medicine lymphoscintigraphy) are routinely used to assess the extent of breast disease and help guide breast interventions.

Limited anatomical studies have been done during the menopausal cycle [2,4,6–8]. Breast ducts are normally cord-like structures with little or no lumen early in the menopausal cycle, and the luminal cells get taller mid-cycle. Ductal cells undergo secretory differentiation, and stromal vascularity increases during the luteal phase of the menopausal cycle. Tritiated thymidine studies show increased labeling of ductolobular units during the luteal phase of the menopausal cycle [9]. With each menopausal cycle in younger women, the number of lobules increases until age 35. During pregnancy, the number of lobular units can increase ten-fold with ductules differentiating into alveoli [10,11].

The rationale for the current study was that a newer modality—low-frequency transmitted ultrasound imaging (volography)—has become available with the capability to differentiate breast tissue types using their speed of sound properties [12–24]. This is an extension of simpler quantitative methods [25]. The term volography refers to the reconstruction by volume rather than slice by slice, as explained in [26], and the volume occupying nature of ultrasound energy, which cannot be constrained to a plane. The 3D nature of the acoustic field requires a 3D model for reconstruction and 3D data acquisition (DA) with a full 2D receiver array as well as a 3D representation of the image. Other groups involved in 3D ultrasound tomography perform slice-by-slice 2D reconstructions, which are concatenated for '3D visualization' of 2D reconstructions. We show in [26–28] that 3D reconstructions are required. This is unique to acoustic imaging, as other modalities can be constrained to or sampled in a plane, and the term 'volography' accurately characterizes this qualitative difference. This method of imaging can carry out accurate volumetric ($\pm 0.2\%$) measurements and also aid in the creation of 3D-printed models of the ductal and glandular systems of a living woman (Figure 1). The technology also enables the measurement of quantitative breast density (% fibroglandular volume, or FGV) in patients [29].

Our null hypotheses are that there is no statistical difference between the variation of pre- and post-menopausal breasts for either glandular or ductal tissue [30,31]. We show this hypothesis is rejected for glandular tissue but not for ductal tissue.

The overall purpose of our breast anatomy research is to determine the role of breast duct physiology in breast cancer formation. To begin this work, we must validate that our methods can determine the changes in breast ductal and glandular tissue anatomy in normal menstruating and post-menopausal women and that they can determine the three-dimensional breast ductal anatomy variation in a cross-section of normal women.

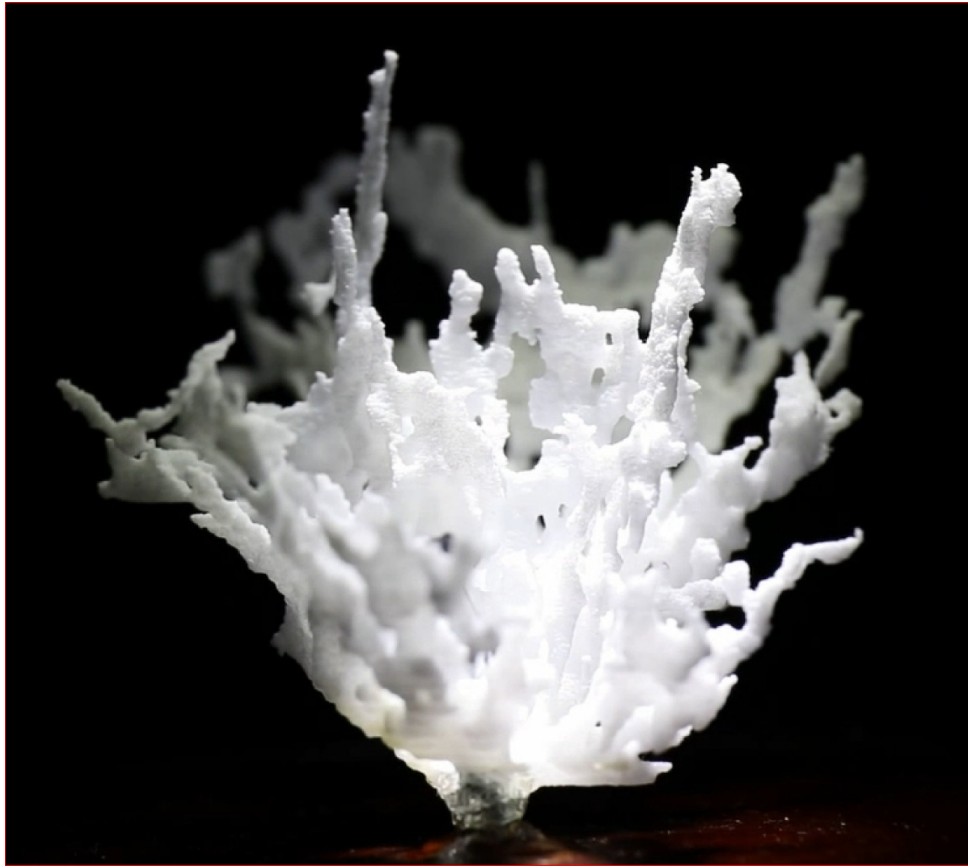

**Figure 1.** A three-dimensional printed model of the ductal system in a living woman. Note the flattened nature of the ducts and the interconnectivity that becomes apparent with high-resolution segmentation.

## 2. Materials and Methods

### 2.1. Study Site, Subjects, and Data Collection

Data were collected at the Marin Breast Health Center using an ultrasonic imaging system (QT Scanner 2000 Model A, QT Imaging Inc., Novato, CA, USA). This is the same low-frequency transmission 3D ultrasound tomography (3D UT) our group has used in clinical settings to identify tissue types [32]. The in-plane resolution is measured at 0.96 and 0.6 mm [28] for reflection and 1.49 mm for transmission. Under IRB approval, we conducted over 400 routine UT breast scans on 24 women (ages 18 to 71), with four (4) of the subjects being post-menopausal (three of those four had complete hysterectomies—ages 71, 66, and 46, and one had no surgery—age 58). These women were scanned weekly, six to ten times, through at least two months of observation. The date of onset of menopause was noted, as well as other relevant data such as age and menopausal history. These cases were further subdivided into three (3) classes: 1. premenopausal: regular menses; 2. perimenopausal: in transition; and 3. post-menopausal: having no menses.

### 2.2. Test Methods

We directly determined automated volumetric breast density and tissue volume calculations derived from speed of sound tissue maps as previously described. For this study, the total breast volume (TBV), FGV, and individual glandular and ductal volumes were calculated for each breast. Ultrasound tomographic volumetric breast tissue calculations were performed using a quantitative three-compartment segmentation algorithm that isolates total fibroglandular breast tissue volumes and glandular and ductal volumes from whole-breast image sets using the speed of sound [33,34]. The method includes an initial separation of the breast from the surrounding water bath, followed by segmentation of the

whole breast into fibroglandular tissue and fat using fuzzy C-mean (FCM) classification. The scan is performed with the breast immersed in a water bath. Because skin and fibroglandular tissue both have relatively high speeds of sound compared with adipose tissue, skin must be segmented out and differentiated from fibroglandular tissue using its proximity to the water's border. The automated software then measures the volumetric breast density using a continuous scale, which has been shown to be more accurate than categorical density scales [35]. Next, we compared the segmented glandular and ductal volumes for all 400 breasts, using the Spearman correlation coefficient (*r*) to quantify the strength of association separately for pre- and post-menopausal subjects. Using the nonparametric Spearman checks for a monotonic—but not necessarily linear—relation. Four (4) null hypotheses related to glandular and ductal tissue variation and the concomitant coefficient of variation (CoV) (listed below) were tested. All statistical analyses were performed using Graphpad Prism® 9 and Excel™ Ver. 2002. Images showing examples of segmentation were shown from the FDA-cleared QT Viewer 2.6.2 (Visualization and Insight Toolkit from Kitware—VTK/ITK 7.1-based). Examples of variations in ductal and glandular tissue over time were visualized using spline interpolation and compared with generic hormonal variations. Glandular and ductal normalized volumes were plotted for all subjects over the course of the data acquisition process, and normalized standard deviations for all 48 breasts in the pre- and post-menopausal groups were shown. Explicit formulae are given to keep the paper self-contained. Comparisons were visualized with scatter plots.

### 2.3. Data Cleaning

Before mathematical statistical analysis, a review of the data showed anomalies that corresponded to either incomplete or poor speed of sound reconstructions. The incomplete or artifacted speed of the sound breast image is the basis for the segmentation, and therefore these results were removed from the data set to avoid contamination. Only two breast scans (out of over 400) were excluded from the analysis.

### 2.4. Data Validation

To establish the general validity of our data, we showed that there is a statistically significant Spearman coefficient between the volumes of glandular and ductal tissue. The Spearman coefficient is derived from a linear regression of the plot of the RANK of ductal tissue vs. the RANK of glandular tissue. This coefficient measures the degree to which there is a monotonic relationship between the ductal and glandular volumes. This relationship need not be linear; any purely monotonic relation will yield a score of 1. We expected on physiological grounds that there will be an approximate monotonic relationship between the two volumes since the same hormones perfuse both types of tissue simultaneously.

We conducted this rank correlation and Spearman coefficient calculation independently for both pre- and post-menopausal breasts.

### 2.5. Glandular vs. Ductal Tissue Behavior over the Menstrual Cycle Compared

The glandular tissue variation for the pre-menopausal breasts was compared to that of the post-menopausal breasts. Similarly, the ductal tissue variation for pre-menopausal breasts was compared to that of post-menopausal breasts. These comparisons are possible since the software algorithm was used to segment glandular vs. ductal tissue, as well as the fibro-glandular tissue in total. The glandular and ductal tissue volume changes were plotted for a representative breast against presumed hormonal changes to validate the correlation of glandular/ductal changes with hormone changes in the breast. The time course of glandular and ductal changes in four breasts was plotted simultaneously to highlight the variability that can occur between breasts (see below).

#### 2.5.1. Null Hypothesis I—Glandular Tissue

The null hypothesis for glandular tissue is that there is no significant difference between glandular tissue volume variance over the course of the study for pre- vs. post-

menopausal women. This hypothesis will be rejected, leading to the intuitive idea that pre-menopausal glandular variance is indeed larger than post-menopausal glandular variation using the *F* statistic.

### 2.5.2. Null Hypothesis II—Ductal Tissue

The null hypothesis for ductal tissue is that there is no significant difference between ductal tissue volume variance over the course of the study for pre- vs. post-menopausal women. It will be seen that this hypothesis cannot be rejected based on the *F* statistic.

### 2.5.3. Null Hypothesis III—Glandular Tissue—Coefficient of Variation

The null hypothesis was that there was no statistically significant difference between the coefficient of variation (CoV) for glandular breast tissues in the breasts of pre- and post-menopausal women. This hypothesis will be rejected below based on the *t*-test for two population means (with an unequal and unknown variance)

### 2.5.4. Null Hypothesis IV—Ductal Tissue—Coefficient of Variation

The null hypothesis is that there was no statistically significant difference between the coefficient of variation (CoV) for ductal breast tissues for pre- and post-menopausal women. This hypothesis cannot be rejected based on the *t*-test calculated below.

Finally, we calculated the unnormalized variance over the study time period of glandular vs. ductal tissue for both pre- and post-menopausal women and saw significant differences in their values.

### *2.6. Coefficient of Variation*

The overall 'coefficient of variation', *CoV*, defined as the ratio of the standard deviation over the average, was plotted for the pre- and post-menopausal cases.

$$CoV \equiv \frac{sd}{avg} = \frac{\sqrt{\sum_k \frac{(x_k - \overline{x})^2}{n-1}}}{\overline{x}}$$

### *2.7. F Statistic to Compare Variance*

The *F* statistic is used to compare variances. The normal distribution is assumed. We have sample sizes of 362 and 62 breast samples for the pre- and post-menopausal cases, respectively, indicating the validity of this assumption.

The calculated *F* statistic $F_{calc} = \frac{s_{pre}^2}{s_{post}^2}$, where $s_j^2$ is the variance for the *j*th sample (either pre- or post-menopausal), is used to establish whether the variation in glandular tissue and ductal tissue behave differently over time and assess their respective different behaviors in pre- and post-menopausal women. The null hypothesis is that the sample variances are the same for pre- and post-menopausal women for both the glandular and ductal tissues. Therefore, two *F* statistics were calculated.

### *2.8. t-Test for Unequal and Unknown Variances*

Since we have established that the variances for the pre- and post-menopausal cases for glandular tissue are statistically different, we utilize the amended Student's *t*-test to compare means for unequal and unknown variances to compare the average over all breasts of the relative variation of the glandular and ductal tissue for each breast (NB, for ductal tissue, the variances are equal but the sample sizes are not).

## 3. Results

Figure 2 shows the speed of sound image of the entire FGV of a breast in axial (craniocaudal) and sagittal (lateral) views. Figures 3 and 4 show the glandular volume on day 1 of the menstrual cycle (Figure 3) and at the beginning of menstruation (Figure 4) in

the same breast. These comparisons show increases in glandular volume at the onset of menstruation.

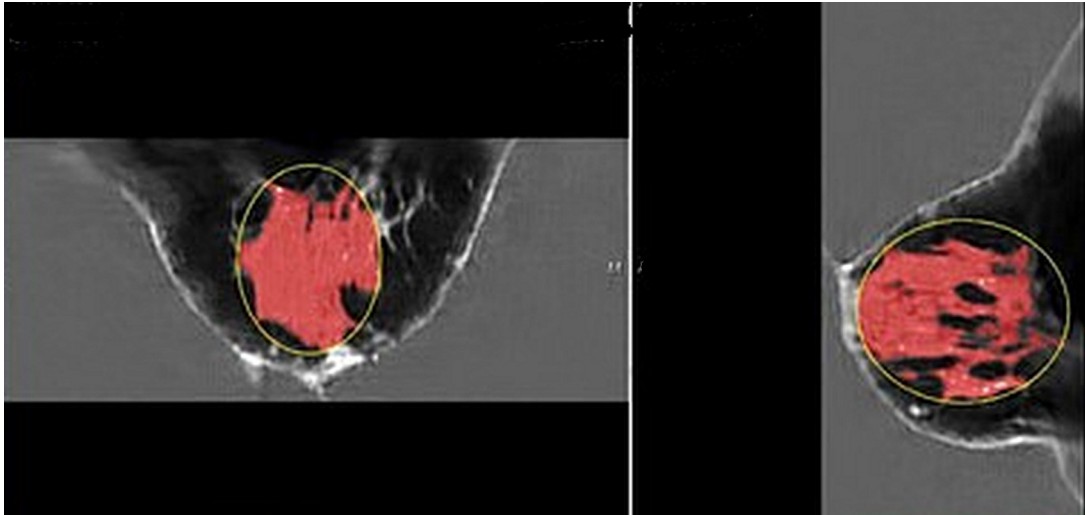

**Figure 2.** Pre-menstrual FGV in the axial (cranial-caudal; (**left**)) and sagittal (lateral; (**right**)) views. The red area is the segmented fibroglandular tissue. The yellow circle shows the initial constraining ellipsoid.

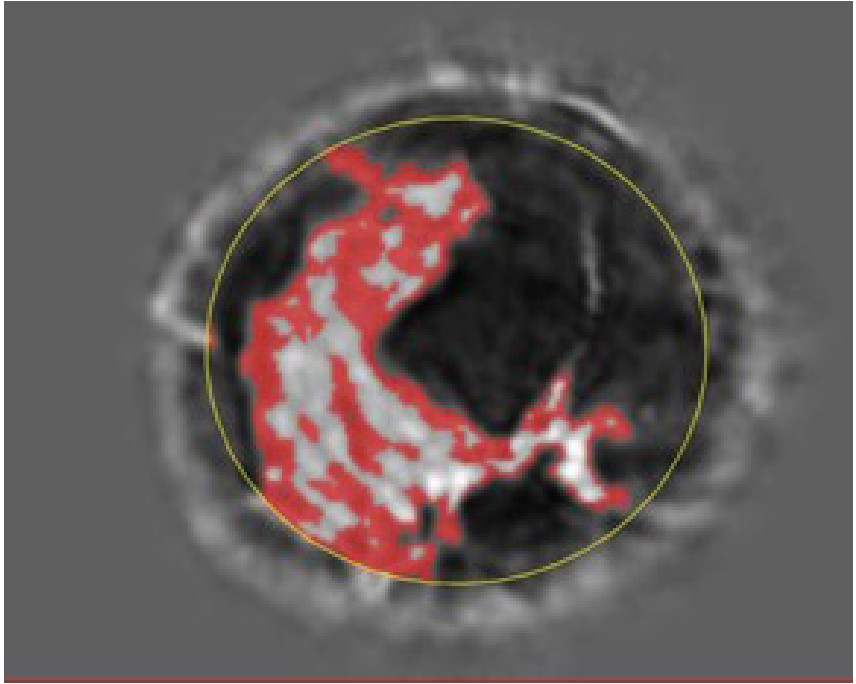

**Figure 3.** Glandular segmentation before menstruation. The red area is segmented glandular tissue. The lighter the grayscale, the higher the speed of sound. Ductal tissue has a higher speed of sound than glandular tissue and is shown in light grayscale outside of the red region. The yellow circle indicates the constraining ellipsoid for the segmentation (also in Figure 4).

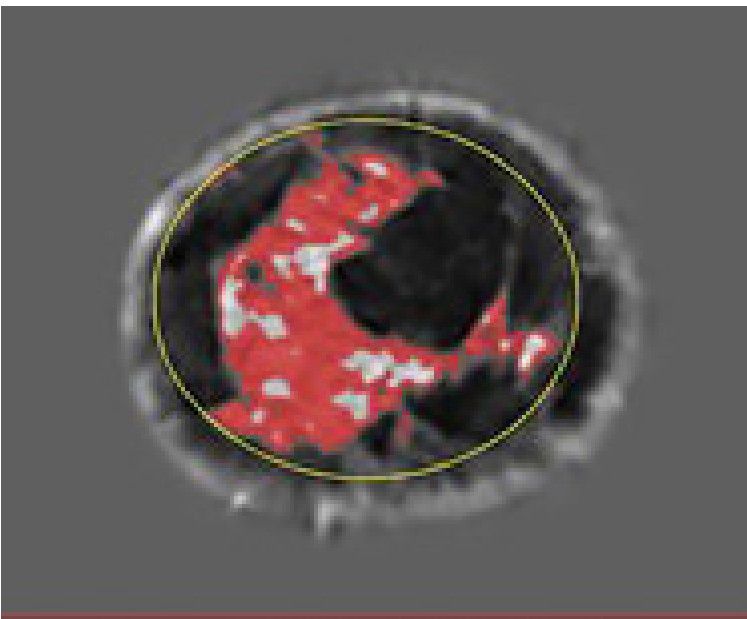

**Figure 4.** Glandular segmentation on day 1 of menstruation in the same breast as in Figure 3 showing an increase in glandular volume. The red area is segmented glandular tissue only. As above, the light gray region represents a higher speed of sound ductal tissue, and the darker region shows fat.

See also the Supplemental Video (online) which shows the 3D-printed ductal volume from the 3D volumetric segmentation of fibroglandular tissue based on QT speed of sound image and other tissue characteristics. As this 3D-printed volume rotates, the topological complexity and relative flatness of the ducts are visible in this post-menopausal woman.

Figure 5 shows the cycling of glandular and ductal volumes during the menstrual cycle in one subject, along with the presumed hormone levels for a typical menopausal cycle. Note that there is an increase in ductal and glandular volumes in the luteal phase—i.e., just prior to menstruation.

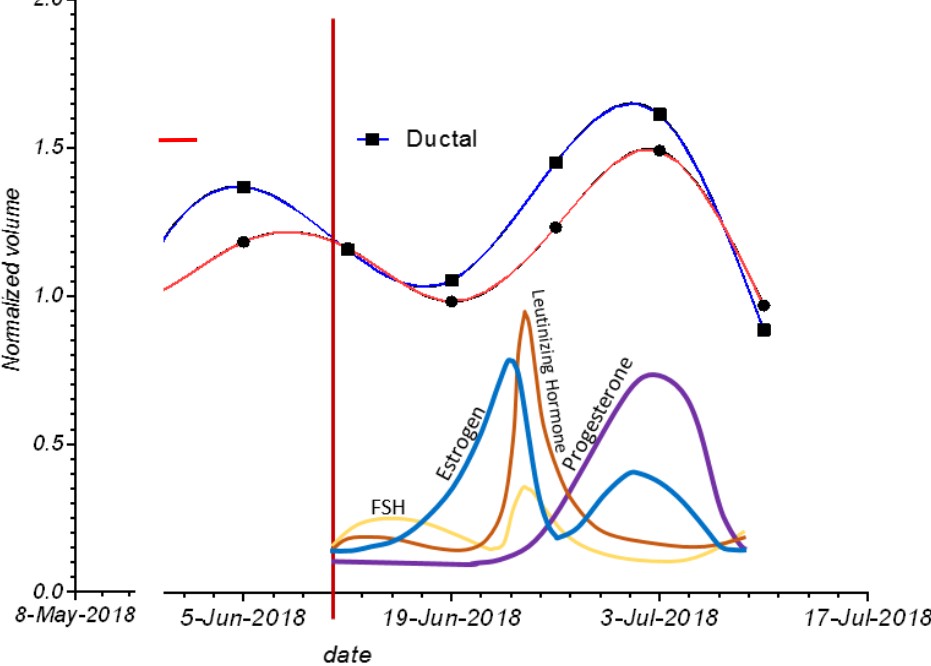

**Figure 5.** The variation of glandular and ductal tissue volumes during the menstrual cycle plotted with the presumed hormone changes. The red vertical line also shows day 1 of menstruation.

Figure 6 shows the changes in ductal and glandular volumes in the breasts of four women during their menstrual cycles. Note that there is inter-subject variation with respect to the exact days in the menstrual cycle when glandular volume increases.

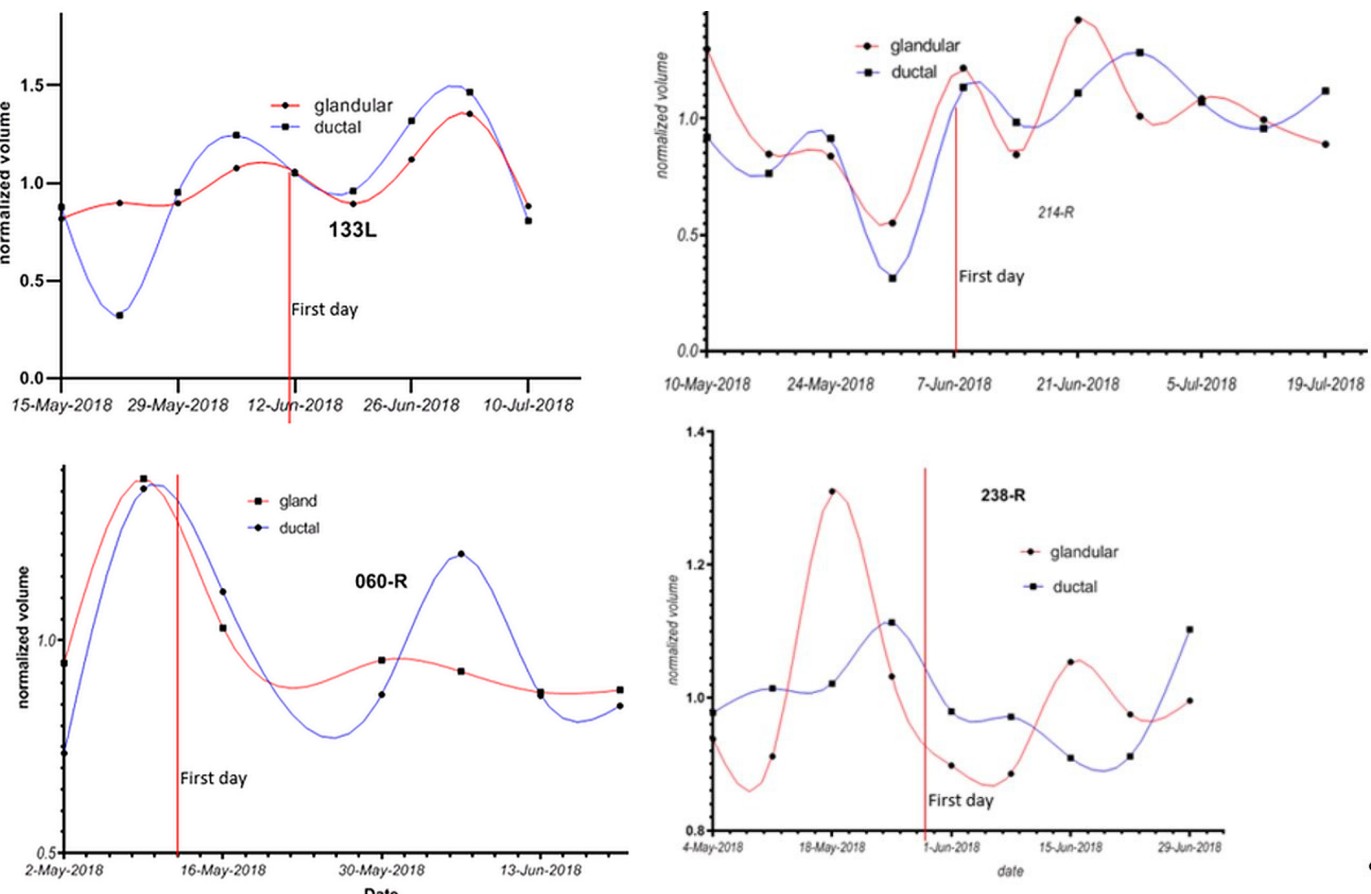

**Figure 6.** Glandular and ductal volumes over time for 4 breasts. The first day of the menstrual cycle is aligned for each of the breasts, showing that there is woman-to-woman variation in the exact days in the luteal cycle where glandular volume increases. Ages: 133L: 30–40 yrs; 214R: 20–30 yrs; 060R: 40–45 yrs; and 238R: 40–45 yrs.

Figure 7 shows the normalized glandular (orange line) and ductal (blue line) volumes in all women during their menopausal cycles. The volumes are divided by the total breast volume.

Note that the ductal and glandular volumes track with one another for individual subjects, although there is variation in the volumes (due to breast size differences) between women.

Figure 8 shows the normalized glandular (orange line) and ductal (blue line) volumes in all four post-menopausal women during their menstrual cycles.

As shown in Figure 7, the ductal and glandular volumes track with one another for individual subjects, but there is variation in the volumes (due to breast size and other tissue related differences) between women.

Figure 9 shows the normalized standard deviation (CoV) for all 48 breasts.

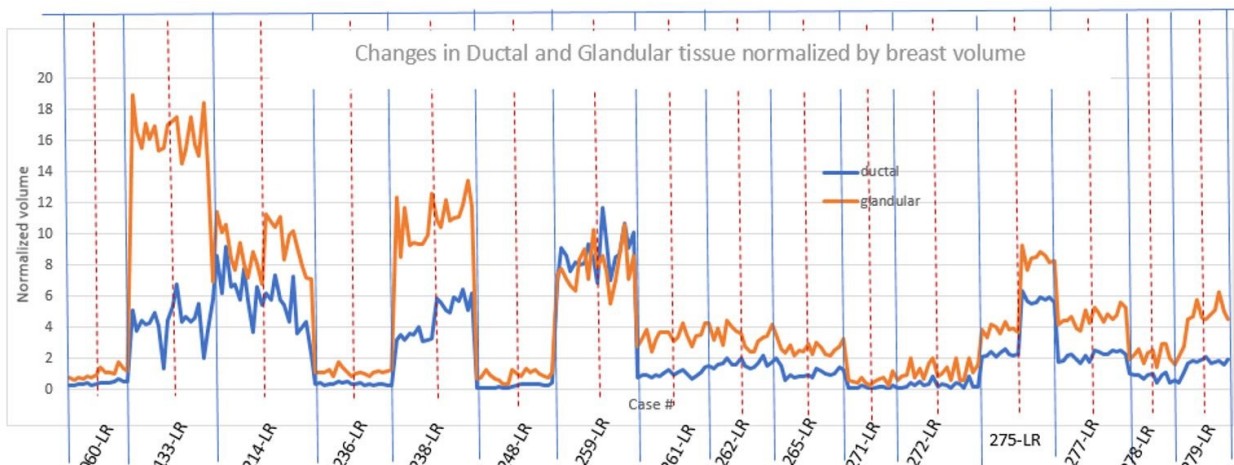

**Figure 7.** For all premenopausal women, the glandular and ductal volumes were divided by the total breast volume and expressed as a percentage. The horizontal axis lists the subject identifiers. Note that each subject has two breasts represented: the first is the left breast, and the second is the right breast for that subject (LR stands for Left, Right). Each breast (L and R) has multiple sample points throughout the study period. The variation within one breast and between breasts is thus shown. Case # = Case number, e.g., 259-LR.

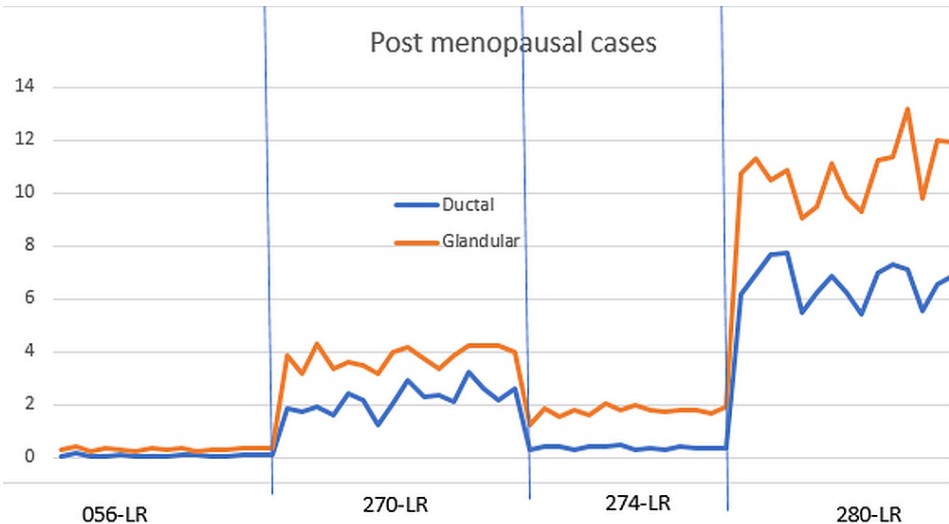

**Figure 8.** Glandular and ductal volumes for post-menopausal cases. The glandular and ductal volumes were divided by the total breast volume to normalize them. LR = left and right breasts.

Note also that the CoV average for the pre-menopausal and post-menopausal ductal tissues is almost identical. These results indicate that there is significantly more variation in glandular tissue during the menstrual cycle in pre-menopausal women than in post-menopausal women. However, it is interesting that post-menopausal women do show small changes in ductal and glandular tissue during each month, suggesting that there are residual cyclic hormonal effects on the breast operating in women throughout their lifetimes [36].

Table 1 shows the percentage changes in glandular and ductal volumes in all breasts during their menstrual cycles normalized to their individual breast volumes.

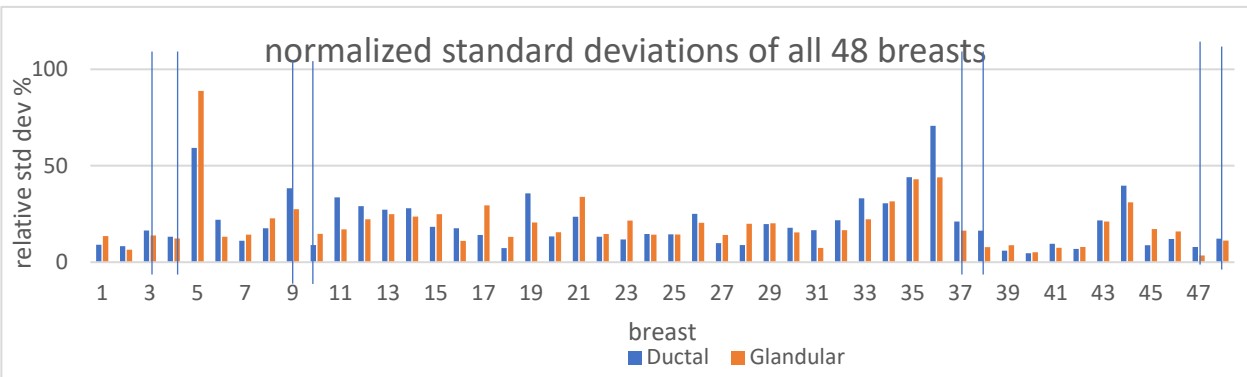

**Figure 9.** Normalized standard deviations for all breasts, also called the coefficient of variation (CoV), are the standard deviations divided by the average glandular/ductal volume over the course of several cycles. Note the variability of the CoV. The vertical lines represent 8 cases (4 subjects—left and right breasts) that are post-menopausal. Note the correlation of CoV for glandular and ductal tissue, giving further confidence in image quality and segmentation.

**Table 1.** Comparison of glandular and ductal variation for pre-menopausal and post-menopausal populations.

| | Pre-Menopausal | | Post-Menopausal | |
|---|---|---|---|---|
| | **Glandular** | **Ductal** | **Glandular** | **Ductal** |
| Avg coefficient of variation | 20.17 (5–43.9)% | 19.64 (2.6–70.6)% | 12.44 (3.44–24)% | 19.84 (2–41%) |
| Avg std dev | 4.35 cc | 1.8 cc | 2.2 cc | 1.5 cc |

The normalized standard deviation (coefficient of variation) as defined above in the Materials and Methods section is clearly scale-invariant and is best expressed as a percentage. It is also referred to as the coefficient of variation (CoV). Note that when the 48 cases are subdivided into 30 pre-menopausal breasts and 8 post-menopausal breasts, the average CoV for the ductal tissue was about the same in pre- and post-menopausal women: 19.64 and 19.84%, respectively. On the other hand, the average CoV for glandular volume was 20.17% for pre-menopausal breasts vs. 12.44% for post-menopausal breasts—a significant difference.

*3.1. Null Hypotheses I and II*

3.1.1. Glandular Tissue Comparison

Glandular tissue has greater variation in pre-menopausal breasts than in post-menopausal breasts.

NB: The $F$ statistic $F_{calc} = \frac{s_1^2}{s_2^2}$ for comparison of variance is calculated to be 4.04 = (4.354/2.167)².

$F$ = 4.04. From one-tailed Fisher tables, the $F$ statistic for 319 *d.f.* over 62 *d.f.* is 1.69 at the 0.01 level of significance.

Since $F_{calc} = 4.04 > 1.69 = F_{319,62;.01}$ we reject the null hypothesis I from the Test Methods section that the variance for pre-menopausal breasts is the same as for post-menopausal breasts for glandular tissue, we accept the alternate hypothesis that the variation for the pre-menopausal breasts is greater than the variation for the post-menopausal breasts. This is a stronger statement than merely accepting the null hypothesis that variation is greater for pre-menopausal breasts than for post-menopausal breasts.

Coefficient of Variation (CoV)

For the glandular tissue note, the average CoV is 20.17% with a range of 5–44% for pre-menopausal breasts vs. 12.44% with a range of 3.44–24% for post-menopausal women.

### 3.1.2. Ductal Tissue Comparison

Ductal tissue does not have significantly more variation in pre-menopausal breasts than in post-menopausal breasts.

The calculated *F* statistic for comparison of variance is calculated to be $1.355 = (1.77/1.519)^2$.

The *F* statistic for 319 *d.f.* over 62 *d.f.* is 1.48 at the 0.05 level of significance.

$F_{calc} = 1.36 < 1.48 = F_{319,62;.05}$, so, for the ductal tissue, we cannot reject the null hypothesis II from the Test Methods section that the variance for ductal tissue for pre- and post-menopausal breasts has equivalent variance.

Coefficient of Variation (CoV)

For the ductal tissue, the average CoV is 19.64% with a range of 2.6–70.6% for pre-menopausal breasts vs. 19.84% with a range of 2–41% for post-menopausal breasts.

### 3.1.3. Total Fibroglandular Tissue

The average increase in fibroglandular tissue volume (as measured by the algorithm) for menstruating women was 25%, and the ranges are shown in Table 1. For post-menopausal women, the average change in breast volume was 19%, and the ranges are shown in Table 1.

When we used segmentation analysis to look further at anatomic detail (segmenting fibroglandular tissue into ducts and glands), one can see that the ductal tissue does not appreciably change in either the pre- or post-menopausal subjects, whereas the glandular tissues change significantly in both situations: pre- and post-menopausal women.

### *3.2. Null Hypotheses III and IV*

#### 3.2.1. *t*-Test for Two Population Means (Variances Unequal and Unknown)

The *t*-test can be applied to the glandular tissue to determine if the coefficient of variation (CoV) is substantially different between pre- and post-menopausal women. The *t*-test for comparison of means with unequal and unknown variances (heteroscedastic) is given by several formulas; the one we used is as follows:

$t_G = \dfrac{\overline{x}_{pre} - \overline{x}_{post}}{\left( \dfrac{s^2_{pre}}{n_{pre}} + \dfrac{s^2_{post}}{n_{post}} \right)^{1/2}}$ with the degrees of freedom given by the following formula:

$$d.f. = \left( \frac{\left( \dfrac{s^2_{pre}}{n_{pre}} + \dfrac{s^2_{post}}{n_{post}} \right)^2}{\left( \dfrac{s^2_{pre}}{n_{pre}} \right)^2 \left( \dfrac{1}{n_{pre}+1} \right) + \left( \dfrac{s^2_{post}}{n_{post}} \right)^2 \left( \dfrac{1}{n_{post}+1} \right)} \right) - 2$$

where the standard formula for variance estimates is used under the assumption that the variances are unequal, as established above. Note that this formula reverts to the standard one when $s_{pre} = s_{post}$ and $n_{pre} = n_{post}$.

#### 3.2.2. Null Hypothesis III—Glandular Tissue CoV

Direct calculation gives *the d.f.* = ~19 and $t_G = 2.656 > t_{19}(\alpha = 0.01) = 2.539$ (from tables).

Therefore, we reject the null hypothesis that the *CoV* for pre-menopausal breasts is the same as the *CoV* for post-menopausal breasts and accept the alternative hypothesis that the average *CoV* for pre-menopausal breasts is greater than the average for post-menopausal breasts.

This *t*-test has shown that the coefficient of variation (*CoV*) for glandular tissue in pre-menopausal breasts is statistically significantly larger than the *CoV* for post-menopausal breasts.

$$CoV^{Gla}_{Pre} >> CoV^{Gla}_{Post}$$

### 3.2.3. Null Hypothesis IV—Ductal Tissue CoV

For the ductal tissue, *d.f.* = ~17 and $t_D = 0.045 < t_{17}(\alpha = 0.01) = 2.567$, so the null hypothesis is not rejected. Note that even with the poor significance level of $a = 0.1$, the null hypothesis cannot be rejected since $t_{17}(\alpha = 0.1) = 1.333$, which is still greater than the calculated *t*-statistic for ductal tissue.

The *t*-test shows that for ductal tissue, the *CoV* for pre-menopausal breasts is not significantly different than for post-menopausal breasts:

$$CoV_{\text{Pre}}^{Duc} \approx CoV_{\text{Post}}^{Duc}$$

Finally, we observe that there is a difference between the variation of tissue in pre-menopausal and post-menopausal women.

*Pre-menopausal subjects*: In particular, there is a statistically larger variance in glandular vs. ductal tissue in premenopausal women since the calculated *F* statistic: $F_{calc} = \frac{s_{Gland}^2}{s_{Duct}^2} = 6.06 > 1.415 = F_{(319,319,.001)}$ is greater than the tabular value for $a = 0.001$ significance.

*Post-menopausal subjects*: On the other hand, the same *F* statistic for post-menopausal women is:

$$F_{calc} = \frac{s_{Gland}^2}{s_{Duct}^2} = 2.035 < 2.2217 = F_{(62,62,.001)}$$

which is less than the tabular value, so there is no evidence that the variance of glandular tissue is larger than the variance of ductal tissue for post-menopausal women.

Thus, we have verified that the relative variation of glandular vs. ductal tissue is quite different for pre-menopausal vs. post-menopausal women in a quantitative manner. We have summarized these results in Table 2 below, where CoV = coefficient of variation = standard deviation/average and pre-/post-M = pre-/post-menopausal.

**Table 2.** Summary of null hypotheses.

| Null Hypothesis | | | |
|---|---|---|---|
| I Glandular tissue | Variance for pre-M = post-M | Reject | Glandular tissue has greater variance in pre-M breasts than in post-M breasts |
| II Ductal tissue | Variance for pre-M = post-M | Accept | Cannot conclude ductal tissue has greater variation pre-M vs. post-M |
| III Glandular tissue | CoV for pre-M = post-M | Reject | The CoV for pre-M breasts is greater than the CoV for post-M breasts |
| IV Ductal tissue | CoV for pre-M = post-M | Accept | Cannot conclude that ductal tissue has a greater CoV pre-M vs. post-M |

### 3.3. Spearman's Coefficient

3.3.1. Ductal Tissue vs. Glandular Tissue Pre-Menopausal Breasts

Figure 10 shows the Spearman coefficient for glandular vs. ductal volumes in pre-menopausal women. The Spearman *r* (*R*) here is 0.853, indicating a strong likelihood of a monotonic relationship between glandular and ductal tissue. As is known, the Spearman *r* is a rank–rank plot that is robust, therefore, against non-normal distributions and the presence of outliers. Since the distribution of volumes is not known to be normal and has not been studied extensively, the Spearman *r* is appropriate here. It indicates a monotonic relation between ductal and glandular volume tissue in pre-menopausal breasts, which can be expected physiologically and supports the accuracy of 3D UT in this context.

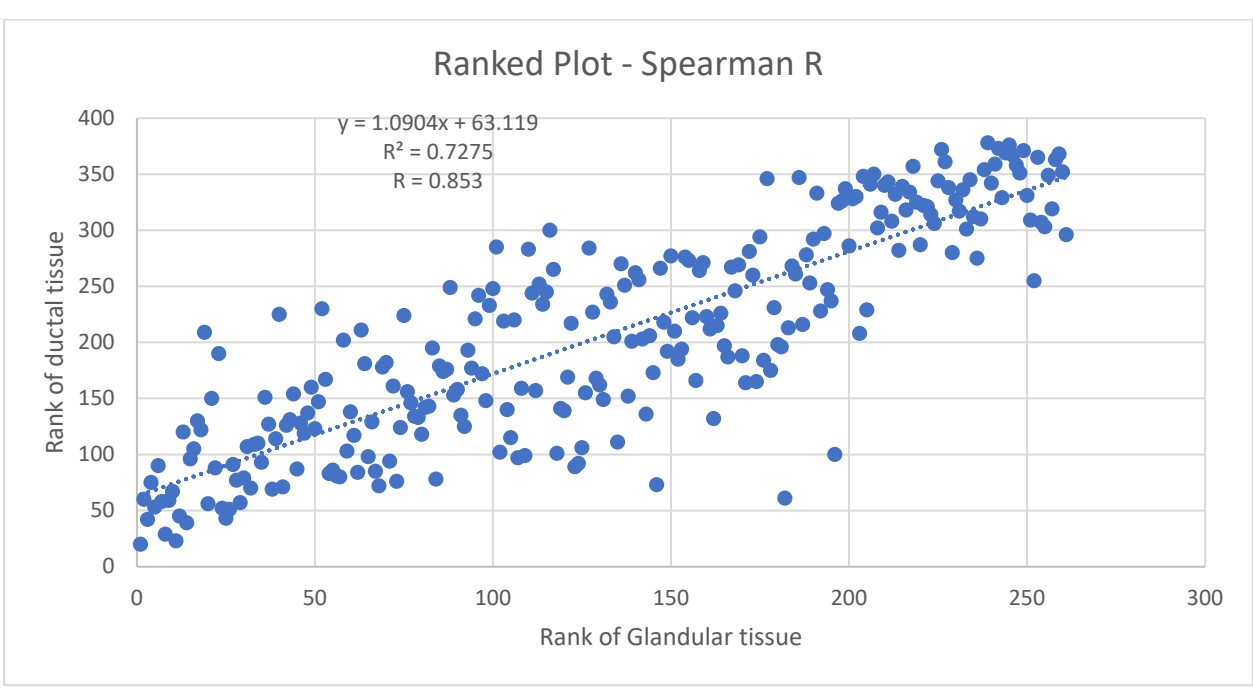

**Figure 10.** Spearman coefficient plot for pre-menopausal breasts. The rank of the ductal tissue plotted against the rank of the glandular tissue.

3.3.2. Ductal vs. Glandular Tissue for Post-Menopausal Breasts

Figure 11 shows the Spearman coefficient for glandular vs. ductal volumes in post-menopausal women. The Spearman R here is 0.8733, indicating a strong likelihood of a monotonic relationship between glandular and ductal tissue.

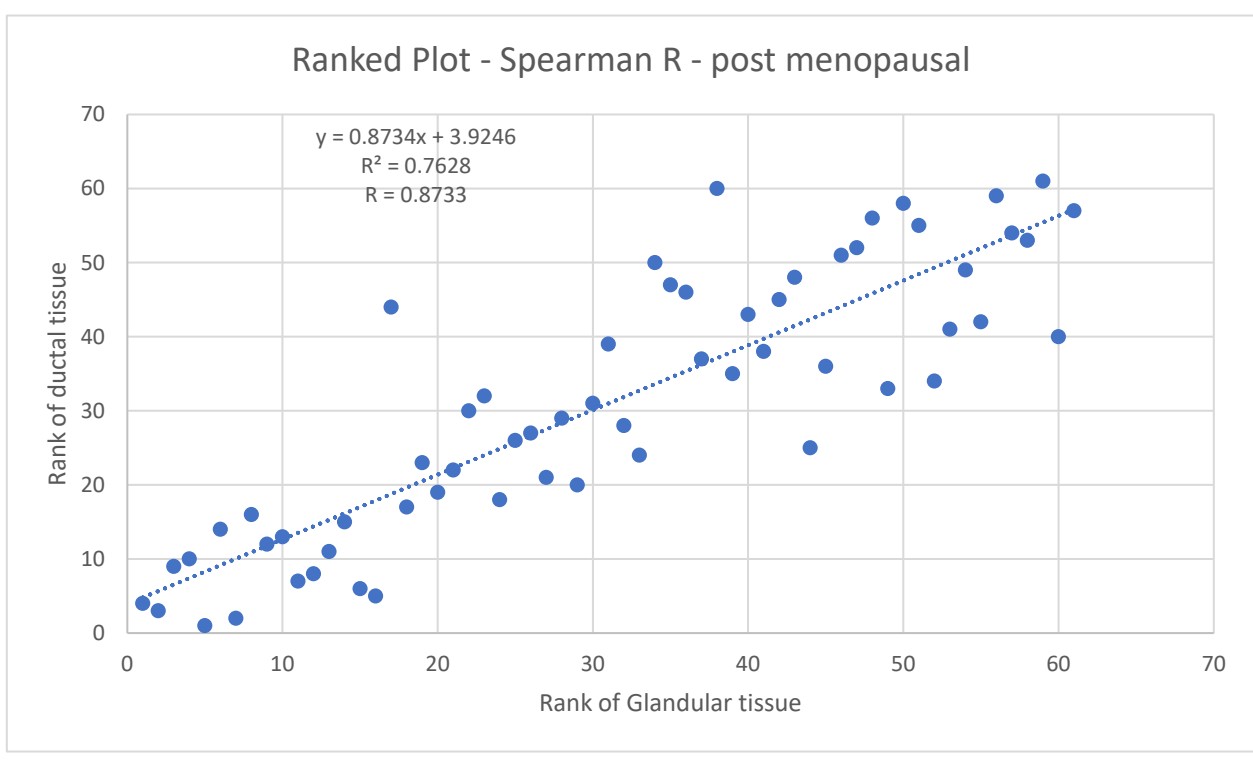

**Figure 11.** Spearman coefficient plot for post-menopausal breasts. Rank of ductal tissue vs. rank of glandular tissue.

These results give us further confidence in the accuracy of our methods since there are physiological reasons to expect such correlations.

### 3.4. Summary of the Null Hypotheses

Glandular tissues of the female breast change significantly during the menstrual cycle—especially in pre-menopausal women's breasts. The ductal tissue does not significantly change during the menstrual cycle in either pre-menopausal or post-menopausal women's breasts.

## 4. Discussion

We establish the quality of our data by removing outliers due to faulty 3D ultrasound reconstruction. Next, we observed a high correlation between the glandular and ductal tissue with a Spearman coefficient of $R = 0.853$ for pre-menopausal breasts and 0.89 for post-menopausal breasts, which is expected for physiological reasons. Note that this high correlation does not contradict our findings of statistically significant differences in the behavior of pre-menopausal and post-menopausal breasts, since the correlations are carried out within the respective sub-divisions of pre- and post-menopausal breasts.

Furthermore, we established using the *F*-statistic that the variance of the glandular tissue behaved significantly differently than the variance of the ductal tissue. In particular, the variances for pre- vs. post-menopausal breasts were statistically different for the glandular tissue, whereas they were not statistically different for the ductal tissue. Finally, we observed that the ductal average coefficient of variation (CoV) for *glandular* tissue decreased from 20.17% to 12.44%, whereas the *ductal* average CoV did not significantly vary between pre-menopausal and post-menopausal breasts. This agrees with the *F-statistic* analysis used to compare the variance behavior of the glandular tissue vs. the ductal tissue from pre- to post-menses breasts.

The glandular tissue showed a larger variation in the pre-menopausal breasts than the ductal tissue.

It is well known that breast density increases by 7–9% during the luteal phase of the menopausal cycle [8]. Chan was able to show a 7% increase in fibroglandular volume in the luteal phase using MRI [37]. Partridge [38], using similar methodology, showed a 6% increase in fibroglandular volume in the luteal phase of the menopausal cycle. Fowler [39] showed that the increased parenchymal volume of the breast during the luteal phase of the menopausal cycle was not due to water content alone. No one has heretofore been able to show differences between ductal and glandular tissue volumes during the menstrual cycles of pre- and post-menopausal women.

This study shows that using a high-resolution imaging method and tissue segmentation, a more accurate description of tissue changes during the menopausal cycle is possible. We modified our data by removing cases (breasts) that were inadequately imaged with our 3D ultrasound algorithm (2 out of more than 400), since this leads to incorrect segmentation. We then performed statistical analyses of the different behavior between glandular and ductal tissue and determined the coefficient of variation (CoV = standard deviation/average) of the glandular vs. ductal tissue over the course of the study for each tissue type: glandular and ductal.

The low-frequency transmitted sound imaging technique has the advantage of gathering approximately 120 GB of data as the armature rotates around the breast during image acquisition. This technique thus provides high resolution image sets with quantitative accuracy ($\pm 0.2\%$ accuracy). This precision, coupled with the ability of the sound speed to identify ductal vs. glandular tissue [13], allows the study of these two tissue types during the menopausal cycle. These results demonstrate that 3D UT suggests that (1) quantitative fibroglandular tissue density (i.e., Quantitative Breast Density: QBD or Fibroglandular Volume: FGV) varies over the course of a woman's menstrual cycle and (2) the glandular tissue has a greater change in volume than the ductal tissue. This group of data from

menstrual study subjects could be useful for working out some of the methodology for the studies of women before, during, and after lactation.

It is not clear why our results have shown larger changes in FGV than previous studies using MRI, although it is possible that MRI is visualizing the breast differently than 3D UT; moreover, the MRI methodology has been manual, although recent work utilizing Dixon sequences and iterative segmentation shows effective automated segmentation [40]. The numbers in this study are larger than those determined by MRI using manual methods: studies using a manual segmentation methodology show breast volume varying by 3% and FGV varying by 6% [38] and 13% [41]. The fifth edition of the BI-RADS lexicon gives recommendations to assess the amount of fibroglandular tissue via MRI and assigns four classes: ACR-MRI-a, b, c, and d. This study supports the conclusion in [40] that objective measures in 3D using MRI or 3D UT appear to be better than subjective estimates.

## 5. Conclusions

In conclusion, we show that low-frequency 3D-transmitted UT (volography) can accurately measure breast ductal and glandular volumes independently in women in vivo, and we have previously shown that this method is comparable to quantitative breast density measurements using other breast imaging modalities, including MRI. We also believe that the high resolution and quantitative accuracy of the full wave inversion used in this study enable a distinction to be observed between glandular and ductal tissue, unlike other modalities. This means that we were able to observe qualitatively different behavior in glandular and ductal tissue in vivo. We believe this represents the first time such an observation has been made. Further work (larger studies, controlling for potentially confounding factors, more diverse demographics, etc.) should be carried out to validate these initial results.

The clinical implications of this study are that low-frequency transmitted UT (volography) may be a useful clinical method for measuring—separately—glandular and ductal breast tissue volumes as a measure of treatment effects in patients. While the QT scanner was used here, the details given in previous publications allow the duplication of these results [27,28], i.e., we emphasize the scientific relevance of these results. Since the method is automated, there is a lower possibility of an interpretive error by the user or clinician when estimating breast tissue volume changes after surgery, radiation therapy, chemotherapy, hormone therapy, or selective hormone modulator therapies. This is important since not all women's breast tissues respond in the same way to treatments, and a non-response suggests that a change in treatment strategy may be indicated. Note also the safety of 3D UT since there is no ionizing radiation, no compression, and no contrast agent. The device is based on ultrasound and is FDA cleared for adjunctive imaging and imaging in dense breasts and young women with risk factors. There have been ~15,000 scans carried out in clinics in the US (CA, NY, UT, AZ, MI, etc.) and ongoing studies to optimize the inclusion of 3D UT into clinical work flow. Finally, the methodology used in this study may be useful for researchers interested in ductal anatomy, ductal fluid diagnostics, or breast cancer treatments via the ductal route.

**Supplementary Materials:** The following supporting information can be downloaded at: https://www.mdpi.com/article/10.3390/tomography10050060/s1: FibroglandularRotate.mp4: showing a 3D-printed ductal volume from the 3D volumetric segmentation of fibroglandular tissue based on QT speed of sound image. As the 3D-printed volume rotates, the topological complexity and relative flatness of the ducts are visible in this post-menopausal woman. This is believed to be the first such 3D-printed volume made from in vivo ultrasound tomography.

**Author Contributions:** Conceptualization, J.K. and S.L.; methodology, J.K. and J.W.; software, J.W.; validation, J.K., J.W. and S.L.; formal analysis J.W.; investigation J.K. and J.W.; resources J.K. and S.L.; data curation, J.W.; writing original draft preparation J.K. and J.W.; writing—review and editing, J.K., J.W. and S.L.; visualization, J.W.; supervision, J.K. and S.L.; project administration, J.K. and S.L.; funding acquisition, S.L. and J.K. All authors have read and agreed to the published version of the manuscript.

**Funding:** This research was funded by Tower Dr. Susan Love Fund for Breast Cancer Research of the Tower Foundation. 8767 Wilshire Blvd, Suite 401, Beverly Hills, CA 90211, ccc@towercancer.org.

**Institutional Review Board Statement:** This study was conducted in accordance with the Declaration of Helsinki and approved by the Institutional Review Board of Western Institutional Review Board (protocol code BR004; WIRB Pro Num: 1167638; date of approval: 25 August 2017).

**Informed Consent Statement:** Informed consent was obtained from all subjects involved in the study.

**Data Availability Statement:** The data presented in this study are available on request from the corresponding author (the data were collected under the auspices of QT Ultrasound® Labs).

**Acknowledgments:** We gratefully acknowledge financial support from the Tower Cancer Research Foundation: Dr. Susan Love Fund for Breast Cancer Research: https://www.towercancer.org/ (accessed on 2 May 2024). We thank N. Pirshafiey for help with the 3D printed volume.

**Conflicts of Interest:** JW and JK have financial connections to QT Ultrasound Labs/QT Imaging, Inc. Dr. Susan Love (deceased) was the founder of the Dr. Susan Love Breast Cancer Research Foundation.

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
