# Peer review of "Breast Glandular and Ductal Volume Changes during the Menstrual Cycle: A Study in 48 Breasts Using Ultralow-Frequency Transmitted Ultrasound Tomography/Volography"

_tomography, doi:10.3390/tomography10050060_

Round 1

Reviewer 1 Report

Comments and Suggestions for Authors

On the whole, this is a very well done study. The capability of 3D UT to separately evaluate different components of breast parenchyma is definitely promising from the perspective of future clinical applications. However, there are a few issues concerning the paper that should be addressed:              

·        In "Materials and Methods", please state the spatial resolution of the UT scans.

·        With regard to your conclusion that 3D UT is equal or even superior to other imaging modalities for breast density assessment, at least when it comes to MRI, it does not seem to be supported by any previously published studies, apart from two considerably old papers that you cited in the last paragraph of the „Discussion“ section. In the current era, MRI provides 3D T1-weighted images of the human breast that are highly correlated with the prevalence of the fibroglandular tissue, and fully automated volumetric measurement approaches are already available (please see reference below). Therefore, an adjustment of both the „Discussion“ and the „Conclusion“ section appears necessary.

Wengert GJ, Helbich TH, Leithner D, Morris EA, Baltzer PAT, Pinker K. Multimodality Imaging of Breast Parenchymal Density and Correlation with Risk Assessment. Curr Breast Cancer Rep. 2019 Mar;11(1):23-33. doi: 10.1007/s12609-019-0302-6. Epub 2019 Jan 17. PMID: 35496471; PMCID: PMC9044508.

Comments on the Quality of English Language

The manuscript is well-written; I have only found minor linguistic errors.

I created an annotated version with suggested corrections.

Author Response

On the whole, this is a very well done study. The capability of 3D UT to separately evaluate different components of breast parenchyma is definitely promising from the perspective of future clinical applications. However, there are a few issues concerning the paper that should be addressed:  

Reply:

We thank the reviewer for the positive comments. We have accepted all the suggestions in their attached document and thank the reviewer for the careful analysis and critical reading of the text.

In "Materials and Methods", please state the spatial resolution of the UT scans.

Reply:

We have placed the resolution cell size in the Materials and Methods section.

With regard to your conclusion that 3D UT is equal or even superior to other imaging modalities for breast density assessment, at least when it comes to MRI, it does not seem to be supported by any previously published studies, apart from two considerably old papers that you cited in the last paragraph of the „Discussion“ section. In the current era, MRI provides 3D T1-weighted images of the human breast that are highly correlated with the prevalence of the fibroglandular tissue, and fully automated volumetric measurement approaches are already available (please see reference below). Therefore, an adjustment of both the „Discussion“ and the „Conclusion“ section appears necessary.

Reply:

We agree with the reviewer that MRI does have supporting evidence that it is capable of measuring fibroglandular tissue (although without differentiating between glandular and ductal tissue). We thank the reviewer for bringing this work/research  to our attention.  We have included the important work of Wengert et al. in the references and commented on this in the discussion and conclusion sections, although we point out the first reference given in Wengert et al. refers to digital breast tomosynthesis not MRI, the rest of the paragraph does present good evidence for the utility of MRI in this regard.

Reviewer 2 Report

Comments and Suggestions for Authors

This is a nice work on investigating breast glandular and ductal volume changes during the menstrual cycle using low frequency 3D transmitted ultrasound tomography (volography).

1.        Since volography is a concept proposed by the authors’ group, it would be helpful for the readers to provide a figure with related descriptions on volography.

2.        The term “volography” appears in the title, but it appears only in the Conclusions section in the text. The readers may not easily understand the term. In addition, the term should be mentioned in the abstract.

3.        Line 29. “in vivo” should be italic.

4.        Line 116. “relation. .” to “relation.”

5.        Line 169 and similar occurances. “t-test”: “t” should be italic, but not “test.”

6.        Line 180. The equation is better defined in a separate line.

7.        Line 186. The equation is better defined in a separate line. “jth”: “j” should be italic.

8.        Line 212. Please check “as in Figure 4,” as this figure is Figure 4.

9.        Line 221. “changes..” to “changes.”

10.    Line 254. “normalized” to “Normalized”

11.    Table 1. “Pre menopausal -” to “Pre-menopausal”

12.    Table 1. “post menopausal” to “Post-menopausal”

13.    Lines 301-302. “Pre and post Menopausal” to “Pre- and post-menopausal”

14.    Line 304. “2.6 – 70.6%” to “2.6% – 70.6%”

15.    Line 305. “2 – 41%” to “2% – 41%”

16.    Line 317. “t-test”: “t” should be italic.

17.    Line 324. “Where” to “where”

18.    Line 365 and similar occurances. “pearman R”: “R” should be italic.

19.    Line 392. Please check the end of this sentence.

Reviewer 3 Report

Comments and Suggestions for Authors

Dear authors,

I read with interest the article “Breast Glandular and Ductal Volume Changes During the Menstrual Cycle: A Study in 48 Breasts Using Ultralow Frequency Transmitted Ultrasound Volography” conducted by James Wiskin et al. I would like to commend the authors for conducting a comprehensive study on the potential of low frequency 3D transmitted ultrasound tomography (3D UT) to differentiate breast tissue types and measure glandular and ductal volumes in vivo. However, there are some major critical points that need to be considered.

Major critical points:

Firstly, the sample size of 24 women, including only four postmenopausal subjects, is relatively small. The study could have been strengthened by including a larger and more diverse sample size to ensure the generalizability of the findings. This limitation should be mentioned in the discussion.

Secondly, the study did not control for the potential confounding factors such as age, body mass index, and hormonal therapy, which could have influenced the results. Future research should consider controlling for these factors to ensure the accuracy of the findings. This fact should be mentioned in the future direction section.

Thirdly, the study did not provide any information on the sensitivity and specificity of 3D UT in differentiating breast tissue types. The authors should consider providing these data to establish the clinical utility of this technique.

Lastly, the study did not provide any information on the feasibility and safety of using 3D UT in a clinical setting. The authors should consider conducting a feasibility study to assess the potential of using this technique in clinical practice. This aspect should be mentioned in the discussion section and in the future directions section.

Minor critical points:

1.     Please indicate the standard deviation in table 1 without the use of the ± symbol.

2.     The Spearman rank correlation coefficient (rho) is a nonparametric measurement of correlation used to determine the relation between two sets of data. The Spearman coefficient is calculated based on the ranks of the data points, rather than their actual values, making it a useful tool for analyzing data with non-normal distributions or outliers. Please address this aspect in the 3.3 section.

In conclusion, while the study provides some promising findings, there are several critical points (major and minor) that need to be addressed before accepting the findings. Further research is needed to establish the clinical utility and feasibility of using 3D UT to differentiate breast tissue types and measure glandular and ductal volumes in vivo.

Comments on the Quality of English Language

Minor revisions to the English language are necessary.

Reviewer 4 Report

Comments and Suggestions for Authors

General Comments

I was impressed with this manuscript in that it presented novel findings on an important topic: the in vivo measurement of breast tissue, looking at the effect of both menstruation and menopause.

There was a good introduction (lines 35 to 42), and I liked the rationale for the study (lines 80 to 84), while I agreed with the bold statement in the Conclusions (lines 438 to 441): "This means that we were able to observe qualitatively different behavior in glandular and ductal tissue in vivo. We believe this represents the first time that such an observation has been validated."

I'm not sure that the term "Volography" in the title is appropriate, even though I am aware this is a made-up word introduced previously by these authors. I think the word "Tomography" is the correct description, as the authors seem to suggest in the first of their Keywords (line 30).

I was interested to see that Dr Susan Love is included as a co-author, even though she passed away on 2 July 2023. This pertains to the statement in Author Contributions (lines 462 to 463) that "All authors have read and agreed to the published version of the manuscript." This implies that the manuscript was prepared prior to Dr Love's passing, which is possible given that the data appear to have been gathered in 2018 (cf. Figures 5 and 6).

I am satisfied with the quality and manner in which the data have been presented in the 11 figures, although I found the presentation of the statistical findings on pages 11 and 12 (lines 279 to 354) quite difficult to follow. I would strongly suggest the authors try to summarise these findings in one or two tables.

Specific Comments

line 168   ... the coefficient of variation (CoV) for glandular ...

line 201   Please ensure that the description of Figures 3 and 4 is correct

line 211   Figure 4 needs to be presented at the same size as Figure 3

line 212   ... in the same breast as in Figure 3 showing ...

Figure 5  The key to the red line is missing the circle and word "Glandular"

line 246   ... volumes for post-menopausal cases.

line 358   ... different for pre-menopausal vs post-menopausal women ...

line 384   ... for pre-menopausal breasts and 0.873 for ...

line 391    ... for pre- vs post-menopausal breasts were ...

line 515    The third author's name is missing. It should be "O'Brien"

line 560    Reference 38 has now been published. It is missing the name of the second author (Bilal Malik) and the published details: 30(11):2674-2685, 2023.

Comments on the Quality of English Language

The quality of the English language in the manuscript is just fine.

Round 2

Reviewer 3 Report

Comments and Suggestions for Authors

Dear authors,

You have responded comprehensively to my review, addressing both the major and minor points raised. You acknowledged the limitations of your study regarding the sample size and lack of diversity, attributing them to the novelty of the technology at the time of data collection. You emphasised the longitudinal nature of their study and highlighted the potential for future research with larger and more diverse cohorts.

Additionally, you recognised the importance of controlling for confounding factors in future studies, such as age, BMI, and hormonal therapy. You also noted the difficulty in obtaining sensitivity and specificity data without a biopsy and highlighted the ongoing need for validation and feasibility studies in a clinical setting.

Also, you made the requested changes, including removing the ± symbol from Table 1 and providing a more accurate description of the Spearman rank correlation coefficient in Section 3.3.

However, despite the thorough response, there's a discrepancy in the iThenticate report, showing a 25% match for the revised manuscript. This suggests that there may still be significant similarities to existing literature that need to be addressed to ensure originality and avoid potential issues of plagiarism. It would be prudent to reevaluate the manuscript to further reduce any similarities and ensure compliance with academic integrity standards.

Author Response

We thank Reviewer 3 for their comments regarding our attempts to comply with the original criticisms of the paper.

Regarding the iThenticate plagiarism note:

We have published fairly extensively in the literature and in presentations/proceedings over the past several decades.

While the subject of this paper is unique and novel (no one has published the evidence to indicate the ability to segment glandular from ductal tissue and follow their structures over several years), the terminology is somewhat peculiar to 3D ultrasound tomography.

We know no one in this area has published these capabilities of 3D UT because no one else has this ability.  We follow the literature in IEEE journals, medical journals, MDPI journals, proceedings of Medical and technical conferences, etc. and are aware of the capabilities in these different research groups and venues. So we are confident these results are unique.

I used iThenticate to verify the uniqueness of the paper and the highest percentage of duplication was 2% and these were in press releases and some recent papers/presentations. The overlap were the name of the company and  “low frequency transmitted 3D ultrasound tomography” or similar phrases that would be impossible to avoid in describing our technology. It is low frequency, and based on transmission ultrasound, it is 3D and it is ultrasound tomography.

It would be impossible for us or other researchers or for a reporting agency to properly describe our technology without referring to it with this terminology.

The rest of the 2% was in the references section, which if the same references were used would be inevitable.

I have changed “low frequency transmitted 3D ultrasound tomography” to “low frequency transmission 3D ultrasound tomography” to avoid the overlap in iThenticate.

I believe there is no plagiarism here.